# Valorization of Olive Mill Byproducts: Recovery of Biophenol Compounds and Application in Animal Feed

**DOI:** 10.3390/plants12173062

**Published:** 2023-08-25

**Authors:** Giulia Francesca Cifuni, Salvatore Claps, Giuseppe Morone, Lucia Sepe, Pasquale Caparra, Cinzia Benincasa, Massimiliano Pellegrino, Enzo Perri

**Affiliations:** 1CREA Research Centre for Animal Production and Aquaculture, 85031 Bella-Muro, Italy; salvatore.claps@crea.gov.it (S.C.); giuseppe.morone@crea.gov.it (G.M.); lucia.sepe@crea.gov.it (L.S.); 2Division of Animal Production, Department of Agriculture, Mediterranean University of Reggio Calabria, 89124 Reggio Calabria, Italy; pasquale.caparra@unirc.it; 3CREA Research Centre for Olive, Fruit and Citrus Crops, 87036 Rende, Italy; cinzia.benincasa@crea.gov.it (C.B.); massimiliano.pellegrino@crea.gov.it (M.P.); enzo.perri@crea.gov.it (E.P.)

**Keywords:** valorization of olive mill wastewater, phenols, circular economy

## Abstract

This study aimed to recover the phenols from olive oil mill wastewater, a major pollutant of the oil industry, by using spray-drying technology to produce a new feed with a nutraceutical value for animal feed supplementation and to evaluate its effect on the productivity and nutritional quality of ewe milk. Forty-five Sarda ewes in late lactation (150 ± 2 d) and with homogeneous live weight (52 ± 1.5 kg) were randomly allotted into three groups and fed with three dietary treatments containing increasing levels of polyphenols: 0% (C), 0.1% (T0.1), and 0.2% (T0.2) of dry matter. No effect of the dietary treatments was found on the milk yield and composition. Interestingly, milk urea content (*p* < 0.0001) and somatic cell counts (*p* < 0.001) decreased as the level of polyphenols inclusion in the diet increased. The inclusion of phenols (0.2% of dry matter) in the diet of sheep was effective in increasing the vaccenic (C18:1 trans-11) and rumenic acid (C18: cis-9 trans-11) levels, which are beneficial for human health. Finally, the recovery of polyphenols via spray-drying technology and their incorporation into a new fortified feed can be a valid strategy for naturally improving the nutritional value of milk while valorizing an oil industry byproduct, reducing environmental impact, and promoting waste reuse that is in line with circular economy principles.

## 1. Introduction

Olive (*Olea europaea*) cultivation plays an important economic and social role in the Mediterranean region. In Europe, 5,028,159 hectares of land are used for the cultivation of the olive tree, with an average annual production of 2,124,009 tons of oil, of which about 75% is produced in Southern Europe [1]. Besides olives and olive oil, the olive agro-industrial sector produces large amounts of olive byproducts, which are considered extremely toxic, with a high negative environmental impact, as well as olive mill residual solids and olive mill wastewater (OMW). OMW is a liquid effluent derived mainly from the water used for the various stages of oil production and vegetable water from the fruit [2] and also contains tannins, lignins, long-chain fatty acids, reduced sugars, proteins, and phenolic compounds which are toxic to micro-organisms and plants. The chemical characteristics and composition of OMW depend on the olive variety, origin, culture conditions, and extraction processes [2,3].

The global production of OMW is estimated to be around 30 × 10^6^ m^3^, with 98% produced in Mediterranean areas [4]. Due to its high chemical oxygen demand (COD 200 kg/m^3^) and biochemical oxygen demand (BOD 100 kg/m^3^), as well as its high concentration of heavy metals, strong acidity, and an important fraction of phenolic components, OMW is considered a critical source of environmental pollution [4]. Conversely, the presence of a high content of polyphenols in this waste matrix increases its potential for enhancement and recovery [5,6].

Recently, the use of OMW has been successfully proposed for different applications, and many studies have focused on obtaining compounds with high added value, i.e., phenolic extracts, through different approaches, including enzymatic and chromatographic techniques, solvent extraction methods, and membrane processes, as well as spray-drying technology [7].

The use of unconventional byproducts that are high in phenolic compounds as a source of bioactive compounds for the feeding and nutrition of small dairy ruminants has recently gained a lot of attention in the zootechnical sector [8,9]. Many studies stated that the moderate levels of polyphenols in the diets of ruminants can enhance animal performance by improving dietary protein utilization and increasing the quality of the derived foods [10,11]. The ability of dietary polyphenols to modulate the rumen bio-hydrogenation of polyunsaturated fatty acids (PUFAs) [12] improves the quality of the lipid fraction of dairy products by increasing the concentration of beneficial fatty acids (PUFAs, vaccenic, and rumenic fatty acids) and enhancing the oxidative stability of the products [13]. In addition, Branciari’s studies [14] have highlighted that dietary supplementation using olive mill wastewater for dairy sheep increases the levels of phenolic compounds, such as tyrosol, hydroxytyrosol, verbascoside, and pinoresinol in the resulting milk and cheese, enhancing their nutraceutical value.

The positive effects of polyphenols on animals’ health have been evidenced by the reduction in intestinal parasites in sheep [15], improvements in the cell-mediated immune response, a reduction in inflammatory processes [16], and improvements in the antioxidant status of animals [17].

The aim of this study was to recover the phenols from oil mill wastewater by using spray-drying technology and produce a new feed with a nutraceutical value for animal feed supplementation and to evaluate their effect on the productivity and nutritional quality of milk from ewes fed with different levels of phenols.

## 2. Results

### 2.1. Ingredients and Chemical Composition of the Diets and OMW

The concentrate offered to the ewes of the T0.1 and T0.2 groups had the same dietary ingredients as the concentrate supplied to the ewes of the control group (C); however, in order to maintain a similar crude protein concentration between treatments, the C group had a different wheat bran content (Table 1). The chemical composition of OMW is shown in Table 1. As reported, the spray-dried olive oil mill wastewater contains a dry matter content of 93%, protein and fat contents of 3.1% and 2.4%, respectively, and as expected, a high oleic acid concentration.

### 2.2. Phenolic Profiles of Spray-Dried Olive Oil Mill Wastewater

The main phenolic compounds found in spray-dried olive oil mill wastewater are oleuropein derivatives (54.67%), tyrosol (17.03%), hydroxytyrosol (12.35%), verbascoside (5.83%), and hydroxytyryloleate (4.70%). Other phenolic compounds comprise caffeic acid, vanillic acid, p-coumaric acid, apigenin, luteolin, diosmetin, rutin, and oleuropein. The total amount of phenols identified corresponds to 11,991 mg/kg. Table 2 shows the phenolic profiles of the supplemented diets. The data are reported in mg/kg.

### 2.3. Nutritional Composition of Milk

The milk yield recorded during the experimental trials was not affected by dietary treatments. The dietary integration of polyphenols that were recovered from olive oil mill wastewater did not affect the fat, protein, lactose, and casein contents (*p* > 0.05), whereas the urea level decreased in the milk from ewes fed with high levels of phenols (*p* < 0.0001). The cell somatic counts were affected by the dietary integration of phenols (*p* < 0.001). Notably, a lower count of somatic cells was found in the milk from the T0.1 and T0.2 groups when compared to the C ones (Table 3).

### 2.4. Milk Fatty Acid Composition

The dietary integration of spray-dried olive mill wastewater significantly affected the proportion of almost all fatty acids. The fatty acid composition of milk from different dietary treatments is shown in Table 4. The levels of short-chain fatty acids (C4:0–C10:0) were similar in all experimental groups. The content of individual branched-chain fatty acid (BCFA) iso-14:0, iso-16:0, ante-iso C17:0, and iso-C18:0 was not influenced by dietary treatments, whereas a higher content of iso-C15:0 in the milk from the control group was found when compared to other groups (Table 1).

In relation to the saturated fatty acids, the levels of C16:0 (*p* < 0.05) and C21:0 (*p* < 0.001) fatty acids decreased in the T0.1 and T0.2 milk samples when compared to those of the C group.

Furthermore, vaccenic acid (C18:1trans11, *p* < 0.001), rumenic acid (C18:2 cis9t11; *p* < 0.05), and elaidic acid (C18:1trans9, *p* < 0.01) were found in a higher proportion in the T0.1 and T0.2 milk in comparison to samples from the control group (C).

The level of C20:4n-6 (*p* < 0.05) increased in the T 0.1 and T0.2 milk samples when compared to the C group. The dietary integration of phenols affected the trans total fatty acids, with a greater proportion found in the milk samples from the T0.1 and T0.2 groups when compared to those of the control, whereas the content of odd-chain fatty acids decreased in the milk from the supplemented group (T0.1 and T0.2; *p* < 0.05).

No effect of dietary treatments was observed for saturated, monounsaturated, polyunsaturated, and the sum of n-3 and n-6 fatty acids. The integration of phenols into the ewes’ diets did not affect the atherogenic (AI) and thrombogenic indexes or the P/S and n-6/n-3 ratios (*p* > 0.05, respectively).

### 2.5. E-Nose Analysis

The LDA plots of the milk odours from different dietary treatments, which were assessed by using PEN 3.5, are shown in Figure 1. The e-nose output data were analysed using linear discriminant analysis (LDA) to evaluate the capability of e-nose PEN 3.5 to discriminate between milk odour profiles as a function of dietary treatments. Two discriminant functions (LD) were obtained that explained 95.24% and 2.67% of the total variance, respectively (Figure 1).

## 3. Discussion

As shown in Table 2, olive mill wastewaters have a high concentration of bioactive compounds with diverse biological properties that can be used as ingredients in the food industry to make functional and nutraceutical foods, as well as in the pharmaceutical industry. The application of spray-drying technologies at low temperatures has allowed us to obtain a product with an optimal phenol composition and, thus, a high biological value.

The increasing interest in phenols found in olive oil byproducts is associated with their biological activities: antioxidant, antiatherogenic, antihepatotoxic, hypoglycemic, anti-inflammatory, antitumor, antiviral, and immunomodulating [18,19,20,21,22]. For instance, oleuropein, hydroxytyrosol, tyrosol, and caffeic acid are considerable scavengers of reactive oxygen and nitrogen species (ROS and RNS) [23]. Hydroxytyrosol also has potential anti-inflammatory properties, reducing pro-inflammatory signalling in human monocytes [24].

The inclusion level of up to 0.2% (of dry matter basis) of spray-dried phenols from olive oil mill wastewater into the ewes’ diet had no effect on milk yield or the fat, protein, lactose, and casein contents (Table 3). The lack of effects from the inclusion of phenols into ewes’ diet on milk production and composition is consistent with results reported by Correddu et al. [8], who stated that the phenols supplementation in small ruminants’ diets on milk production and composition did not provide univocal results.

These contrasting results are most likely due to ruminant species, the amount or kind of polyphenol source employed, and the associative effects between the phenol source and other dietary ingredients [25]. Interestingly, milk urea content decreases as the level of phenols included in the diet increases; these data suggest that dietary phenols interact with protein metabolism in the rumen, limiting their ruminal degradation [11].

The effects of dietary phenols on the rumen degradability of proteins, which determines the reduction in milk urea concentration, have been previously reported by Toral et al. [26] and Buccioni et al. [27]. Similarly, Nudda et al. [28] reported that the phenols binding to proteins in the rumen might decrease ruminal protein degradability, ruminal NH3-N production, and, consequently, milk urea excretion, implying their potential role in ruminant nutrition in terms of improving nitrogen utilization and reducing nitrogen excretion in the environment.

In line with earlier studies [29,30], the supplementation of phenols extracted from olive oil mill wastewater in the diets of ewes resulted in a considerable reduction in milk somatic cell counts (Table 3).

As suggested by Min et al. [31], the effect of phenols on somatic cell counts can be explained by the action of secondary metabolites with potential bactericidal action in the mammary gland, which inhibits the proliferation of the most important pathogens for the mammary gland, such as *Staphylococcus Aureus*, *Escherichia coli*, and *Klebsiella pneumoniae.*

The lipid content of ruminant-derived foods, particularly their fatty acid (FA) composition, is influenced by rumen microflora activity and metabolism [12]. The inclusion of phenols in animal diets can modulate rumen micro-organism activities [32]. As a result, many studies have been conducted to investigate the feasibility of modulating rumen microbiota using dietary phenols to improve the fatty acid profiles of foods [26,33], thereby increasing their content of polyunsaturated (PUFA), vaccenic, and conjugated linoleic (CLA) fatty acids, all of which are beneficial to human health.

As expected, the fatty acid profile of milk fat changed as a result of the supplementation from spray-dried phenols recovered from olive oil mill wastewater. The inclusion of phenols (960 mg of spray-dried olive oil mill in each kg of the diet, 0.2% of dry matter) in the diet of sheep was effective at increasing vaccenic (C18:1 trans11) and rumenic acid (C18: 2cis9 trans11, the main isomers of CLA) levels.

These results agree with the studies of Valenti et al. [34], who reported that dietary pomegranate pulp (4.7 g of total phenol in 1 kg of dry matter) enriched the goat’s milk with healthy fatty acids, such as rumenic and vaccenic acids. On the contrary, Capucci et al. [35] stated that the supplementation of dairy ewes’ diets with different doses of phenolic concentrate from olive mill wastewater (0.6 and 1.2 g/kg of dietary dry matter) did not affect the CLA and vaccenic concentrations. Hence, the inconsistency of the results may be related to the amounts of phenols administered and to the ratio of the single compounds present in the extract. The negative or beneficial effects of phenols on the bio-hydrogenation of dietary PUFAs can vary according to animal species, the composition of the basal diet, the source of the phenols, and the amount of their inclusion in the diet [8]. Moreover, in vitro studies by Khiaosa-Ard et al. [36] underlined that the dietary phenols inhibited the last step of the rumen bio-hydrogenation of linolenic and linoleic fatty acids, determining an increase in C18:1 *trans*-11, a precursor for the synthesis of C18:2 *cis*-9, *trans*-11, according to the Δ^9^-desaturase activity in the mammary gland [37]. C18:2 *cis*-9, *trans*-*11* is the main CLA isomer, which has purported anticarcinogenic, anti-atherogenic, anti-inflammatory, and positive immune modulatory properties [38].

According to Khiaosa-Ard et al. [36], the content of total trans fatty acids found in milk fat from ewes fed with phenols was higher when compared to the control group, and this might be due to a slower rate of ruminal bio-hydrogenation, which increased primarily vaccenic acid (C18:1 trans11) and rumenic acid (C18:2 cis9 trans11).

When considering the PUFA contents, according to Cappucci et al. [35], dietary polyphenols increase the level of arachidonic acid (C20:4n-6).

Milk odd- and branched-chain fatty acids are considered biomarkers of rumen function and have been proposed as diagnostic tools to predict shifts in the microbial population associated with diet variation [39]. In our studies, dietary phenol supplementation produced a decrease in the total odd-chain fatty acids in the milk samples from the supplemented groups, suggesting a negative effect of dietary phenols on the rumen bacterial population, which is the main producer of these fatty acids in the rumen [40].

The FA profile of dairy fat is important for the nutritional quality of dairy products. The P/S and n-6/n-3 PUFA ratios, the atherogenic (AI), thrombogenic (TI), and hypocholesterolemic/hypercholesterolemic (H/H) are commonly used to assess the nutritional value and consumer health of animal fat [41]. In general, a PUFA-to-SFA ratio of above 0.45 and n-6-to-n-3 ratio of below 4.0 are required in the diet to prevent cardiovascular diseases [42]. In the present study, the P/S and the n-6/n-3 ratios in all experimental groups were considerably lower than the recommended values. The atherogenic and thrombogenic indexes take into account the different effects that single fatty acids might have on human health. It is supposed that milk fat with high AI and TI values may contribute more easily to the development of atherosclerosis or coronary thrombosis. The range of values obtained for AI (2.66–2.88) agreed with results reported in the literature for ewe milk [41].

As reported in Figure 1, by mimicking the activity of the human nose, an electronic nose can play an important role in the objective discrimination of milk origin. The LD algorithm clearly separated three distinct groups in relation to the different dietary treatments; this suggests that the dietary inclusion of polyphenols determines the different patterns of odour in milk.

## 4. Materials and Methods

### 4.1. Experimental Design, Treatment, and Sample Collection

The experimental design was prepared in accordance with Directive 2010/63/EU of the European Parliament [43], which deals with the protection of animals used for scientific purposes.

The experiment was performed on a commercial farm located in the southern region of Italy. Briefly, 45 pluriparous Sarda ewes in late lactation (150 ± 2 d) with homogeneous live weight (52 ± 1.5 kg) were randomly allotted into three groups and were confined in multiple pens. The experimental trial lasted 4 weeks, taking place after 1 week of adaptation to the experimental diet. In the experimental period, the animals received three different diets: the control group (C) was fed with a pelleted concentrate, whereas the experimental group (T0.1) received pelleted concentrate supplemented with the polyphenols recovered from oil mill wastewater by spray-drying at 0.1% of dry matter basis to reach a total concentration of 480 mg/kg of phenolic compounds, and the T0.2 group received a concentrate supplemented with the phenol compounds at 0.2% of a dry matter to reach a total concentration of 960 mg/kg of phenolic compounds. All animals received polyphyte hay ad libitum, and a formulated concentrate was offered daily for a total of 0.4 kg/head. All concentrate ingredients and dry-sprayed phenols were incorporated into the pellets. All dairy ewes consumed the whole daily amount of feed supplied because of the fixed amount of diet provided. The ingredients and chemical composition of the experimental diets are reported in Table 1.

The ewes were milked twice daily, and the daily milk yield was individually recorded. The milk samples, for a total of 180 milk samples (45 animals × 4 sampling days), were collected each week and transported (refrigerated) (4 ± 2 °C) to the laboratory for the physical-chemical analysis, whereas the milk samples for fatty acid composition determination were stored at −80 °C until the analyses.

### 4.2. Sampling Olive Oil Mill Wastewater

During the crop year 2018–2019, olive mill wastewater (OMW) was collected from the oil mill “La Molazza” (Corigliano Calabro, Italy). OMW was collected after centrifugation in a three-phase mill for the olives of Carolea and Dolce di Rossano cultivars and stored in appropriate containers at 8 °C for three days. Subsequently, the OMW was treated by applying the technology of spray-drying by the company “EVRA S.r.L.” (Lauria, Italy) with a “De Lazzari 5G” spray drier (De Lazzaris.r.l., Busto Arsizio, VA, Italy) set at the following operational parameters: inlet temperature: 165 °C, outlet temperature: <80 °C, feeding: 4.5 L/h, and turbine: 2700× *g*. The spray-drying process involves the atomization of OMW into droplets by spraying, followed by the rapid evaporation of the sprayed droplets into a solid powder by a flow of hot air. Before the drying procedure, OMW was treated with maize maltodextrin (40–60%) to achieve a 30% dry residue. The evaporation of moisture from the drops and the formation of dry granules occurs under a controlled temperature (100 °C) and airflow. The moisture content of the final product was 7% [7].

### 4.3. Spray-Dried Olive Mill Wastewater Phenols Analysis

The phenolic compounds were analysed, as described by Benincasa et al. [7]. The extraction of phenols was carried out in an ultrasonic bath in darkness, and 20 gr of the dried olive mill was added to 20 mL of a solution of methanol and water (*v*/*v* 80:20). The separation was obtained by centrifugation at 5000 rpm/min, and the supernatant was recovered.

The remaining residue was re-extracted, as mentioned above, two more times. The resulting supernatants were then combined and analysed by high-pressure liquid chromatography-tandem mass spectrometry (HPLC-MS/MS) using an HPLC 1200 series instrument (Agilent Technologies, Santa Clara, CA, USA) interfaced to an API 4000 Q-Trap mass spectrometer (Applied Biosystem/MSD SCIEX, Concord, ON, Canada), set in a negative multiple reaction monitoring mode (MRM), equipped with a column (Eclipse XDB-C8-A HPLC, Agilent Technologies, 5 μm particle size, 150 mm length, and 4.6 mm i.d.) for chromatographic separation. All reagents (LC/MS grade) and standards for the phenolic assay were purchased from Sigma-Aldrich (Riedel-de Haën, Laborchemikalien, Seelze, Germany) and Extrasynthese (Nord B.P 62 69726 Genay Cedex, France). For the quantification assay, external calibration curves with a least-squares linear regression analysis were used. The calibration curve’s correlation coefficients were between 0.9997 and 0.9999. The amounts of oleuropein and ligstroside derivatives were estimated considering the peak regions of oleuropein glycoside because of a lack of commercially available standards.

### 4.4. Chemical Analysis of Milk and Feed

The milk samples were analysed for fat, lactose, protein, and urea contents using infrared spectroscopy (Milkoscan 6000 FT (Foss Electric, Hillerød, Denmark), and the somatic cell counts were evaluated according to ISO 13366-2/IDF 148-2 [44] using a Fossmatic 5000 (Foss Electric). The diet samples, relative to the dietary treatments, were collected daily, pooled weekly, and stored at −20 °C until further analysis. Table 1 shows the chemical composition of experimental diets and spray-dried olive oil mill wastewater. According to AOAC [45] methods, the dry matter (DM), ether extract (EE), and ash contents of the diets were calculated. The Van Soest et al. [46] procedure was used to assess the fibre fractions, including NDF, ADF, and ADL. The analysis of the fatty acid composition of dietary treatments was carried out using the method of Gray et al. [47].

### 4.5. Milk Fatty Acid Composition

Milk lipids were extracted, as reported by Cifuni et al. [48]. Duplicates of a 10 mL chloroform extract containing 100 mg of fat were methylated by adding 1 mL hexane and 0.05 mL 2 N methanolic KOH [49].

Gas chromatographic analysis was performed on a Varian model 3800 GC instrument (Agilent Technologies, Santa Clara, CA, USA) equipped with an automatic sampler (CP 8410). The FAME content was determined using a fused silica capillary column covered with 100% cyanopropyl polysiloxane (DB 23, J & W), 60 m, 0.25 mm (i.d.), and 0.25 μm film thickness.

The following operating parameters were used: a 1.2 mL/min helium flow rate, a 250 °C FID detector, a 230 °C split-split less injector with a split ratio of 1:100, and a 1 mL injection volume. The temperature programme of the column was as follows: 5 min at 60 °C, then increased at a rate of 14 °C/min to 165 °C, followed by an increase of 2 °C/min to 225 °C, and then held for 20 min [44]. By comparing the retention times to those of the known mixtures of standard fatty acids (Supelco FAME mix37 47885U, Sigma-Aldrich, St. Louis, MO, USA), the individual fatty acid peaks were recognised. Larodan (Malmö, Sweden) provided the individual standard references for CLA isomers (cis-9, trans-11 97%, and trans-10, cis-12 3%). Fatty acids were expressed as a per cent of total fatty acids. The Ulbright and Southgate equations [50] were used to determine the atherogenic (AI) and thrombogenic (TI) indexes.

### 4.6. E-Nose Analysis

The odour fingerprint analysis on milk was evaluated using an electronic nose (PEN3, AIRSENSE Analytics GmbH, Schwerin, Germany), which contains 10 different metal oxide semiconductor (MOS) sensors and is composed of a sensor array system to provide selectivity towards volatile compound classes. A total of 15 mL of milk sample was placed in a 50 mL headspace bottle and kept at 30 °C for 60 min before detection. The flushing time was 150 s, the acquisition time was 150 s, and the sampling interval was 1 s. The data acquisition and processing of the electronic nose data was performed using the workstation software Winmuster 1.6.2.18. Each sample was repeated eight times, and the data with stable measurement results were selected for statistical analysis. The performance description and the gas sensitivity range of the 10 metal oxide sensors are shown in Table 5.

### 4.7. Statistical Analysis

The data were processed by analysis of variance (SAS version 9.3, SAS Institute, Cary, NC, USA) with dietary treatment as a factor and the sampling time as a repeated measure; the means values were compared by Fisher’s LSD test. The data obtained from the sensor array of the electronic nose were analysed by linear discriminant analysis (LDA).

## 5. Conclusions

The main finding of this study is that olive aqueous waste is a source of bioactive and natural phenols. The addition of spray-dried phenols, recovered from olive oil mill wastewater, into the diets of lactating ewes enhanced the nutritional value of their milk by increasing the levels of rumenic and vaccenic fatty acids, which are beneficial for human health. Additionally, the inclusion of phenols in the diet of ewes could be a potential strategy to reduce the milk count of somatic cells; therefore, this field needs further studies.

Thus, the production of new feed for animal nutrition, enriched with natural phenols derived from oil mill wastewater, can be a valid strategy to naturally improve the nutritional value of milk and to valorise the byproducts of the oil industry, reducing the environmental impact and promoting the reuse of waste in accordance with the principles of the circular economy.

## Figures and Tables

**Figure 1 plants-12-03062-f001:**
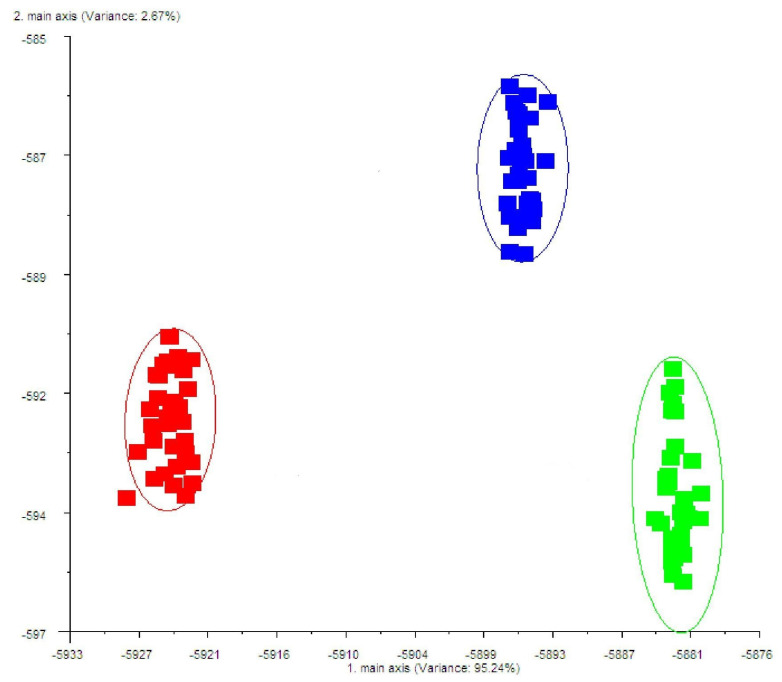
Linear discriminant analysis (LDA) of milk samples from ewes fed with concentrate (C; blue data points), supplemented phenols at 0.1% of dry matter (T0.1; green data point), and supplemented phenols at 0.2% of dry matter (T0.2; red data point).

**Table 1 plants-12-03062-t001:** Ingredients (% as fed basis), chemical composition (g/100 g), fatty acids (g/100 g of total fat), and polyphenolic compounds of diets offered and spray-dried olive oil mill wastewater.

	Diets			
	C	T0.1	T0.2	OMW
Maize grain	27.5	27.5	27.5	
Wheat bran	20.2	20.1	20	
Maize flour	8	8	8	
Maize dried distillers’ grain	8	8	8	
Hulled sunflower seed flour	8	8	8	
Barley grain	8	8	8	
Soybean meal	7	7	7	
Molasses	5.8	5.8	5.8	
Dried beet pulp	3.9	3.9	3.9	
Spray-dried olive mill wastewater phenolics	*-*	0.1	0.2	
Calcium carbonate	1.47	1.47	1.47	
Magnesium oxide	0.43	0.43	0.43	
Sodium bicarbonate	0.94	0.94	0.94	
Sodium chloride	0.47	0.47	0.47	
Vitamin-mineral supplement	0.29	0.29	0.29	
*Chemical composition*				
Dry matter	85.11	85.84	85.84	93
Crude protein	16.52	16.51	16.50	3.1
Ether extract	3.50	3.50	3.50	2.4
Ash	6.50	6.52	6.54	8.5
NDF	25.36	25.31	25.26	9.50
ADF	10.58	10.57	10.55	4.25
ADL	2.35	2.35	2.34	-
*Fatty acids (g/100 g of fatty acids)*				
C16:0	9.45	9.45	9.44	16.5
C18:0	3.36	3.37	3.37	2.5
C18:1 n-9	18.13	18.17	18.21	62
C18:2 n-6	43.85	43.81	43.78	16
C18:3 n-3	2.10	2.10	2.10	0.45
Others	23.11	23.11	23.11	2.55

C = control diet; T0.1 = diet with phenols with 0.1% of dry matter; T0.2 diet with phenols with 0.2% of dry matter; OMW = spray-dried olive oil mill wastewater.

**Table 2 plants-12-03062-t002:** Phenolic profiles of supplemented diets (mg/kg).

	Diets ^1^	
Phenolic Compound	T0.1	T0.2
Catecol	0.245	0.490
Tyrosol	81.715	163.429
Vanillin	1.108	2.216
Hydroxytyrosol	59.236	118.472
Hydroxytyryloleate	22.559	45.117
p-Coumaric acid	0.200	0.401
Caffeic acid	0.116	0.231
Apigenin	0.382	0.764
Luteolin	2.495	4.990
Diosmetin	0.143	0.286
Luteolin-7-O-glucoside	3.542	7.084
Luteolin-4-O-glucoside	0.459	0.918
Oleuropein	4.120	8.239
Oleuropein derivatives	262.223	524.446
Ligstroside derivatives	11.159	22.319
Rutin	1.941	3.881
Verbascoside	27.998	55.996
Sum of phenols	479.640	959.280

^1^ T0.1 = diet with phenols at 0.1% of dry matter; T0.2 diet with phenols at 0.2% of dry matter.

**Table 3 plants-12-03062-t003:** Effect of dietary treatments on milk chemical characteristics.

	Diets ^1^				
	C	T0.1	T0.2	SEM ^2^	*p*-Value
Milk yield, g/d	445	455	448	0.030	0.965
Fat, g/100 g	5.82	5.61	5.54	0.154	0.729
Protein, g/100 g	4.51	4.44	4.33	0.251	0.164
Lactose, g/100 g	4.34	4.42	4.40	0.041	0.378
Casein, g/100 g	3.411	3.357	3.282	0.131	0.7874
Urea, mg/100 mL	51.239 ^A^	52.617 ^A^	43.177 ^B^	1.534	<0.0001
SCC ^3^, 10^3^ cells/mL	2163 ^A^	1012 ^AB^	879 ^B^	313	0.00082

The means in the same row with different letters indicate significant differences ^(A, B^ = *p* < 0.0001). ^1^ C = control diet; T0.1= diet with phenols at 0.1% of dry matter; T0.2 diet with phenols at 0.2% of dry matter; ^2^ SEM = standard error of the means; ^3^ SCC = somatic count cells.

**Table 4 plants-12-03062-t004:** Effect of the dietary treatments on fatty acid composition (g/100 g of the total fatty acids) of ewe milk.

	Diets ^1^				
	C	T0.1	T0.2	SEM ^2^	*p*-Value
C4	3.059	2.748	2.574	0.174	0.159
C6	2.898	2.747	2.669	0.161	0.599
C8	2.971 ^ab^	2.543 ^b^	3.146 ^a^	0.162	0.044
C10	8.682	9.073	9.512	0.428	0.400
C11	0.288	0.253	0.254	0.005	0.742
C12:0	4.085	4.235	4.561	0.174	0.168
C12:1	0.128	0.127	0.138	0.008	0.609
C13:0	0.100	0.0092	0.090	0.005	0.354
C14:0 iso	0.178	0.165	0.167	0.011	0.708
C14:0	10.479	10.908	10.721	0.184	0.279
C14:1 trans	0.155 ^B^	0.200 ^A^	0.211 ^A^	0.009	0.0008
C14:1 cis	0.414	0.419	0.430	0.025	0.877
C15:0 iso	0.587 ^a^	0.506 ^b^	0.511ab	0.027	0.008
C15:0	1.176	1.052	1.061	0.047	0.149
C15:1	0.352	0.303	0.317	0.019	0.215
C16:0 iso	0.089	0.100	0.088	0.006	0.381
C16:0	26.819 ^a^	25.141 ^b^	25.044 ^b^	0.513	0.039
C16:1 trans	0.305	0.324	0.319	0.013	0.593
C16:1 cis	0.623	0.574	0.690	0.112	0.950
C17:0 ante iso	0.552	0.502	0.491	0.044	0.588
C17:0	0.491	0.418	0.427	0.025	0.100
C17:1	0.667	0.609	0.602	0.04	0.413
C18:0 iso	0.237	0.231	0.230	0.02	0.943
C18:0	9.233	9.546	9.570	0.029	0.677
C18:1 t9	0.075 ^B^	0.100 ^A^	0.096 ^A^	0.003	0.0002
C18:1 t-11	0.526 ^B^	0.833 ^A^	0.848 ^A^	0.007	0.0001
C18:1 n-9	18.538	19.635	19.926	0.505	0.317
C18:1 n-7	0.508	0.428	0.431	0.025	0.059
C18:2 t-9t-12	0.284	0.286	0.318	0.019	0.424
C18:2 t-9c-12	0.042	0.079	0.052	0.026	0.582
C18:2 c-9t-12	0.437	0.525	0.516	0.038	0.232
C18:2 n-6	2.438	2.600	2.491	0.083	0.394
C18:3 n-6	0.106	0.107	0.103	0.010	0.956
C18:3 n-3	0.495	0.421	0.413	0.60	0.069
C18:2 c-9t-11	0.690 ^b^	0.809 ^a^	0.750 ^a^	0.022	0.040
C20:0	0.297	0.336	0.317	0.020	0.434
C20:1	0.052	0.061	0.058	0.005	0.549
C21:0	0.081 ^A^	0.036 ^B^	0.050 ^B^	0.005	0.0001
C20:2 n-6	0.061	0.059	0.058	0.005	0.906
C20:3 n-6	0.069	0.073	0.071	0.006	0.872
C20:4 n-6	0.048 ^B^	0.051 ^A^	0.052 ^A^	0.007	0.0022
C20:3 n-3	0.159	0.188	0.178	0.016	0.446
C20:5 n-3	0.053	0.053	0.052	0.019	0.503
C22:0	0.158	0.175	0.161	0.011	0.536
C22:2 n-6	0.053	0.057	0.057	0.005	0.878
C22:4 n-6	0.051	0.056	0.052	0.004	0.767
C22:5 n-3	0.108	0.106	0.096	0.008	0.528
C22:6 n-3	0.083	0.089	0.082	0.006	0.760
SFA ^3^	72.470	70.815	71.666	0.586	0.161
MUFA ^4^	22.347	23.620	22.990	0.525	0.252
PUFA ^5^	5.182	5.564	5.343	0.144	0.197
n-3 ^6^	0.900	0.857	0.822	0.031	0.232
n-6 ^7^	2.828	3.004	2.884	0.083	0.328
P/S ^8^	0.071	0.079	0.075	0.002	0.138
Total trans	1.345 ^B^	1.746 ^A^	1.793 ^A^	0.022	0.001
Branched-chain	1.645	1.505	1.496	0.09	0.463
Odd-chain	2.138 ^A^	1.853 ^B^	1.894 ^B^	0.074	0.027
n-6/n-3	3.148	3.519	3.535	0.150	0.069
AI ^9^	2.803	2.667	2.733	0.095	0.611
TI ^10^	2.987	2.827	2.900	0.071	0.307

Means in the same row with different letters indicate significant differences (^a, b^ = *p* < 0.05; ^A, B^ = *p* < 0.0001). ^1^ C = Control diet, T0.1 = diet with phenols at 0.1% of dry matter, and T0.2 = diet with phenols at 0.2% of dry matter. ^2^ SEM = standard error of the means; ^3^ SFA, saturated fatty acids; ^4^ MUFA, monounsaturated fatty acids; ^5^ PUFA, polyunsaturated fatty acid; ^6^ n-3 = (C18:3 n-3 + C20:2 n-3 + C20:3 n-3 + C20:5 n-3 + C22:5 n-3 + C22:6 n-3); ^7^ n-6 = (C18:2 n-6 + C18:3 n-6 + C20:2 n-6 + C20:3 n-6 + C20:4 n-6 + C22:4 n-6); ^8^ P/S polyunsaturated fatty acids/saturated fatty acids; ^9^ AI, atherogenic index = C12:0 + 4 × C14:0 + C16:0)/(n-3 + n-6 + MUFA); ^10^ TI, thrombogenic index = (C14: 0 + C16: 0 + C18:0)/((0.5 × MUFA) + (0.5 × n-6)+ (3 × n-3) + (n-3/n-6)).

**Table 5 plants-12-03062-t005:** Sensor sensitivities and detection limits for the PEN3 sensor array.

Sensor Number	Sensor Name ^1^	Sensor Sensitives	Detection Limits
1	W1C	Aromatic organic compounds	Toluene, 10 mg kg^−1^
2	W5S	Very sensitive, broad range sensitivity, reacts to nitrogen oxides, very sensitive, with negative signal	NO^2^, 1 mg kg^−1^
3	W3C	Ammonia, also used as sensor for aromatic compounds	Benzene, 10 mg kg^−1^
4	W6S	Detects mainly hydrogen gas	H2, 0.1 mg kg^−1^
5	W5C	Alkanes, aromatic compounds, and nonpolar organic compounds	Propane, 1 mg kg^−1^
6	W1S	Sensitive to methane. Broad range of organic compounds detected	CH3, 100 mg kg^−1^
7	W1W	Detects inorganic sulphur compounds, e.g., H2S. Also sensitive to many terpenes and sulphur-containing organic compounds	H2S, 1 mg kg^−1^
8	W2S	Detects alcohol, partially sensitive to aromatic compounds, broad range	CO, 100 mg kg^−1^
9	W2W	Aromatic compounds, inorganic sulphur and organic compounds	H2S, 1 mg kg^−1^
10	W3S	Reacts to high concentrations (>100 mg kg^−1^) of methane and aliphatic organic compounds	Not determined

^1^ As reported in the “sensors options” of the e-nose software (Winmuster 1.6.2.5, Airsense Analytics GmbH, Schwerin, Germany.

## Data Availability

The authors confirm that the data supporting the findings of this study are available on request.

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
