# Peer review of "Valorization of Olive Mill Byproducts: Recovery of Biophenol Compounds and Application in Animal Feed"

_plants, 2023, doi:10.3390/plants12173062_

Round 1

Reviewer 1 Report

We consider the manuscript very interesting and pertinent to the readers of this Journal’s Special Issue of “Update on the Olive Tree Cultivation: Sustainable Innovative Techniques and Mitigation Strategies against Climate Change”; Section of Crop Physiology and Crop Production.

This study aimed to increase the productivity and nutritional value of ewe´s milk using recovered polyphenols from oil mill wastewater as animal feed supplementation.

This article has a good organization of the contents and the objectives of this work are clearly stated and comprehensively justified.

The introduction briefly covers old and new references and perfectly integrates the theme's main aspects.

Regarding the discussion of the results, we found it suitable and supported with the appropriate correspondent statistical analysis to support the conclusions, which were carefully elaborated and meticulously presented.

Regarding the discussion of the results, we found it suitable. The nice graphic pics/graphs with proper statistical support accompanying the discussion increase the understanding of the discussed theme and clarifies the reasoning. Though, the quality of the Figures ought to be increased (throughout the manuscript) to match the higher standards of the Journal.

The limitations of the study are lacking.

It should be noted that the authors make the data supporting the findings of this study available on request, contributing to greater transparency in science. We congratulate the authors on this!!

Specific Comments:

#1_ L244-247_ Please improve the quality of the picture.

#2_ Please check the name of the microorganisms in italics, e.g. L_287.

#3_ Please check for uniformity in the physical units concerning “-1” vs “/”

#4_Please check for typos such as the formatting with Bold type of some references´ years (e.g. Lines 634, 583, 594, 564.

Author Response

see attach file

Reviewer 2 Report

Dear authors

Congratulations on your interesting work.

Please consider:

line 142- found, instead of founds

line 146 - identifed, instead of identifies

line 287 - pneumoniae, instead of pneumonia

Microorganisms name, should be in italics.

Please consider:

line 142- found, instead of founds

line 146 - identifed, instead of identifies

line 287 - pneumoniae, instead of pneumonia

Microorganisms name, should be in italics.

Author Response

See attach file

Reviewer 3 Report

You'll find remaquers, suggestions, and comments in the attached file.

Author Response

See attach file
